# Mechanical Properties of PALF/Kevlar-Reinforced Unsaturated Polyester Hybrid Composite Laminates

**DOI:** 10.3390/polym14122468

**Published:** 2022-06-17

**Authors:** Siti Nadia Mohd Bakhori, Mohamad Zaki Hassan, Noremylia Mohd Bakhori, Ahmad Rashedi, Roslina Mohammad, Mohd Yusof Md Daud, Sa’ardin Abdul Aziz, Faizir Ramlie, Anil Kumar, Naveen J

**Affiliations:** 1Razak Faculty of Technology and Informatics, Universiti Teknologi Malaysia, Jalan Sultan Yahya Petra, Kuala Lumpur 54100, Malaysia; snadiabakhori@gmail.com (S.N.M.B.); mzaki.kl@utm.my (M.Z.H.); noremyliamb@gmail.com (N.M.B.); mroslina.kl@utm.my (R.M.); yusof.kl@utm.my (M.Y.M.D.); saa.kl@utm.my (S.A.A.); faizir.kl@utm.my (F.R.); 2College of Engineering, IT & Environment, Charles Darwin University, Casuarina, Northern Territory 0810, Australia; 3Kamla Nehru Institute of Technology, Sultanpur 228118, India; anilk@knit.ac.in; 4School of Mechanical Engineering, Vellore Institute of Technology, Vellore 632014, India; naveen.j@vit.ac.in

**Keywords:** PALF, kevlar, hybrid composite laminates, natural fiber, synthetic fiber

## Abstract

Natural and synthetic fibres are in high demand due to their superior properties. Natural fibres are less expensive and lighter as compared to synthetic fibres. Synthetic fibres have drawn much attention, especially for their outstanding properties, such as durability, and stability. The hybridisation between natural and synthetic fibres composite are considered as an alternative to improve the current properties of natural and synthetic fibres. Therefore, this study aimed to determine the physical and mechanical properties of pineapple leaf fibre (PALF) and Kevlar reinforced unsaturated polyester (UP) hybrid composites. The PALF/Kevlar hybrid composites were fabricated by using hand layup method utilising unsaturated polyester as the matrix. These composites were laid up to various laminated configurations, such as [PKP]s, [PPK]s, [KPP]s, [KKP]s, [PPP]s and [KKK]s, whereby PALF denoted as P and Kevlar denoted as K. Next, they were cut into size and dimensions according to standards. Initially, the density of PALF/Kevlar reinforced unsaturated polyester were evaluated. The highest density result was obtained from [KKK]s, however, the density of hybrid composites was closely indistinguishable. Next, moisture absorption behaviour and its effects on the PALF/Kevlar reinforced unsaturated polyester were investigated. The water absorption studies showed that the hybridisation between all PALF and Kevlar specimens absorbed moisture drastically at the beginning of the moisture absorption test and the percentage of moisture uptake increased with the volume fraction of PALF in the samples. The tensile test indicated that all specimens exhibited nonlinear stress-strain behaviour and shown a pseudo-ductility behaviour. [KKP]s and [KPK]s hybrid composites showed the highest tensile strength and modulus. The flexural test showed that [KPK]s had the highest flexural strength of 164.0 MPa and [KKP]s had the highest flexural modulus of 12.6 GPa. In terms of the impact strength and resistance, [KKP]s outperformed the composite laminates. According to SEM scans, the hybrid composites demonstrated a stronger interfacial adhesion between the fibres and matrix than pure PALF composite.

## 1. Introduction

Natural fibres often derived from renewable sources consist of plants, animals, and minerals [1]. Plant fibres are made up of cellulose, which is the most prevalent natural fibre in the world [2,3]. Khan et al. listed the most important plant fibres including abaca, henequen, jute, kenaf, oil, pineapple, ramie, sisal, bamboo, banana, coir, cotton, curaua, flax, hemp, and wood [4]. Animal fibres consist of proteins, such as wool, silk and hair, while mineral fibres, such as asbestos [5]. According to Ortiz et al., natural fibres have a significant benefit over the traditional reinforcing or synthetic fibres [6]. Natural fibres are useful in both economically and environmentally. Furthermore, mechanical and physical qualities of these natural fibres were excellent. Many novel and improved materials, including composites, have resulted from advances in material science. Owing to the rapid growth of the manufacturing sector, materials with greater strength, stiffness, density and cost-effectiveness are in demand as well as improved sustainability [7]. Composite materials are one of these materials that possess these characteristics.

Pineapple leaf fibre (PALF) is one example of natural fibres that showed outstanding fibre qualities, such as high in cellulose, low in cost and eco-friendly with good fibre strength [8]. PALF is made up of cellulose (70–82%), lignin (5–12%), and ash (less than 2%). (1.1%), and have excellent mechanical qualities [9]. PALF has a tensile strength of 413 MPa–6127 MPa and a tensile modulus of 60 GPa–82 GPa [10]. PALF’s outstanding mechanical qualities can be attributed to its high alpha-cellulose content, low microfibrillar angle (14) and superior stiffness and strength [11] as compared to other cellulose-based composite materials. The most well-known synthetic fibres used in the composite industry are Kevlar fibres, glass fibres, and carbon fibres. Kevlar fibre such as Kevlar 49 has a high stiffness, whereas Kevlar 29 has a low stiffness, lightweight, robust, and tough, making them ideal for aeronautical applications [12].

The hybrid composite polymer has gained a lot of attention. Hybrid composites are made by weaving two or more different types of fibre together in a typical matrix. Alternately, reinforcing filler material is introduced into a single matrix or two matrices to form hybrid composites [13]. These provided designers with a selection of wide range of fibre material and geometrical options to meet the environmental requirements and increase its cost effectiveness. Because some reinforcements are excessively expensive, the goal of material hybridisation in composite engineering, nowadays is to create a new material that keeps the benefits of its elements at a lower cost [14]. In the past few years, scientists and researchers have investigated the integration of natural fibres with synthetic fibres in hybrid composites to conserve human health and the environment. Natural fibre-reinforced composites are relatively strong, lightweight, environmentally benign, biodegradable, and have a high potential as building materials [15]. However, they have several disadvantages over synthetic fibres, e.g. carbon and glass fibres, including low flexural modulus, poor moisture resistance to absorption, and reduced strength [16]. In natural fibre-reinforced composites, the poor adhesion between the fibre and matrix is also a concern [17]. As a result, by combining the natural fibres with synthetic fibres of high strength properties can increase the composites’ strength and stiffness, moisture absorption resistance, and, most importantly, balancing the impact and performance of composite materials on the environment [13]. In fact, natural fibres could improve multiple properties when combined with synthetic fibres, potentially lowering the cost of hybrid composites [18]. The high-performance of synthetic fibres properties are critical in the manufacturing industries. These fibres are noted for their high performance, stability, and durability, as well as their long lifespan. 

Previous research on PALF hybrid composite laminates has demonstrated that composite materials have outstanding mechanical properties, as indicated in Table 1. In this instance, it is important to keep investigating into the possibilities of composite materials, specifically cellulosic fibre-based composites. PALF/Kevlar reinforced unsaturated polyester (UP) hybrid composites’ physical and mechanical qualities have yet to be investigated. As a result, the purpose of this research is to assess the physical and mechanical properties of PALF/Kevlar fibre-reinforced UP hybrid composites by using various fibre stacking configurations. In addition, the hybrid laminate failure properties were also investigated.

## 2. Materials and Methods

### 2.1. Materials

Dupont Kevlar yarn in a 2 × 2 twill weave was purchased from Easy Composite Ltd. (Stoke-on-Trent, United Kingdom). The fibre thickness was approximately 0.47 mm. The plain woven PALF was obtained from Mecha Solve Engineering Trading (Kangar, Perlis, Malaysia). Alsey Kimia (M) Sdn Bhd (Seremban, Negeri Sembilan, Malaysia) supplied the unsaturated polyester (UP).

### 2.2. Fabrication of Sample Hybrid Composite

The specimens fabricated were inter-ply non-hybrid and hybrid PALF/Kevlar reinforced UP hybrid composites. Six different types of composites, including two non-hybrid laminates and four hybrid laminates, were produced. After applying the mould release wax to the glass mould surface, peel ply and other fibres were inserted one at a time in the desired stacking sequences ([PKP]s, [PPK]s, [KPP]s, [KKP]s, [PPP]s and [KKK]s). The excess epoxy from roller action dripped down the edges of the specimen and frequently adhered to the glass mould surface. The purpose of using peel ply was to make it easy to remove the final prepared specimen from the glass mould surface. It created a rough surface between the fibre skin layers and the glass mould surface. It also prevented the glass mould from damage when the final composites were extracted from the glass mould surface. The upper mold was then clamped together with the lower plates with the same number of clamp rotations to generate an equivalent thickness of fibre laminate, which was then left to cure for 24 h at room temperature. After the curing process, both composites were removed from the mould. Finally, the specimens were cut into required sizes, and five samples for each parameter were prepared. Specimen properties of the samples were measured by using digital calipers, whereby the average values were taken and tabulated. The properties of the samples fabricated by hand layup technique are tabulated in Table 2. 

### 2.3. Evaluation of PALF/Kevlar Unsaturated Polyester Hybrid Composites

#### 2.3.1. Density Measurement

The Archimedean principle was used to calculate the density. The solid condition of the finally cured composite sample was weighed in air, then weighed again in a known density liquid (alcohol) as shown in Figure 1. The density measurement values in g/cm^3^.

#### 2.3.2. Water Absorption Characteristics

The ASTM D570 standard was used to measure water absorption characteristics. The sample was oven dried for 2 h at 80 °C before measuring the weight of composites (CMH Ltd., Lancing, UK). Five replicates of composite coupons with dimensions of 20 mm × 20 mm × 3 mm were soaked in distilled water. The samples were taken out of the damp environment and dried clean with a microfibre cloth. At every 2 h of water immersion, the moisture uptake statistics were recorded by using a weight balance (EMS 300-3). Equation (1) was used to determine the moisture absorption characteristic [29]:(1)$$\%W=\frac{W_{t}-W_{0}}{W_{0}}\times 100$$

$W_{0}$ is the weight of the dry sample, and $W_{t}$ is the weight of the sample at a recorded immersion time. Fickian’s theory was used to examine the kinetic and diffusion mechanisms. Its relationship is:(2)$$\log\left[\frac{M_{t}}{M_{m}}\right]=\log\left(k\right)+n\log\left(t\right)$$
where $M_{t}$ and $M_{m}$ are the water absorption at time t and saturation point, respectively; k and n are constants.

A crucial parameter in Fick’s model is the diffusion coefficient, or the capacity of moisture to permeate into the molecule of a composite. The initial linear part of the moisture absorption percentage versus the square root of the time curve yields the D value. The diffusivity of the specimen can be calculated by using the equation below [20]:(3)$$D_{z}=\pi {\left[\frac{h}{4M_{m}}\right]}^{2}{\left[\frac{M_{2}-M_{1}}{\sqrt{t_{2}}-\sqrt{t_{1}}}\right]}^{2}$$
where *h* is plate thickness; *M*_2_ and *M*_1_ are moisture content at time *t*_1_ and *t*_2_, respectively; and $M_{m}$ is the weight of saturated specimen.

The following equation was used to predict the moisture absorption behavior of the composite specimens using the theoretical Fick’s second diffusion law [20]:(4)$$M\left(T,t\right)=\left(M_{m}-M_{i}\right)\left(1-\exp\left(-7.3{\left(\frac{D_{z}t}{h^{2}}\right)}^{0.75}\right)\right)+M_{i}$$

#### 2.3.3. Tensile Testing

According to ASTM D3039, a standard rectangular shaped composite laminated specimen was created (d). The specimen had dimensions of 250 mm × 25 mm and a nominal thickness of 1 mm. The tensile test was performed at a crosshead displacement of 1 mm per minute up to breaking. During the testing, an extensometer with the original gauge length, GL = 50 mm, was placed on the specimen to record the sample’s elongation. To obtain consistent results, the experiment was replicated five times using five different samples.

#### 2.3.4. Flexural Test

The ASTM D790-03 standard was used to test the flexural characteristics of the produced PALF/Kevlar UP hybrid composites. The tests were carried out by using a universal testing machine (Shimadzu AG-X Plus, Kyoto, Japan), which had a span to depth of 12.7 mm and a thickness of 150 mm. As illustrated in Figure 2, the flexural testing speed was set at 1 mm/min with a load cell of 20 kN.

#### 2.3.5. Charpy Impact Test

The produced PALF/Kevlar UP hybrid composite was put through an unnotched Charpy impact test to determine its impact characteristics. A pendulum impact tester was used to examine the sample with dimensions of 63 mm × 12.7 mm × thickness. At a release angle of 160°, the pendulum was released with a 4 J energy capacity. The investigation was performed according to the ASTM D256 test standards. Six PALF/Kevlar UP hybrid composite samples were tested to obtain their average impact strength.

#### 2.3.6. Surface Morphology

A scanning electron microscope (SEM; Hitachi TM3000, Tokyo, Japan) was used to analyse the fracture surface morphologies of the PALF/Kevlar hybrid laminate composites. Before scanning, the fragmented areas of the samples were chopped and platinum was evenly deposited over the surfaces. Scanning pictures were obtained at magnifications of 100 and 500 at accelerating voltages of 3 kV–5 kV.

## 3. Results and Discussion

### 3.1. Physical Properties of PALF/Kevlar-Reinforced Unsaturated Polyester Hybrid Composites

#### 3.1.1. Density

The density of PALF/Kevlar reinforced UP hybrid composites that were subjected to various laminated configurations ([PKP]s, [PPK]s, [KPP]s, [KKP]s, [PPP]s and [KKK]s) are as illustrated in Figure 3. In general, a single reinforcement composite shows a higher density as compared to hybrid laminates. The highest density result was obtained from [KKK]s, however, the density of hybrid composites was closely indistinguishable. This result was mainly ascribed to the same number of woven fabric plies resultant in nearly identical in the density of the composites. In addition, as compared to Kevlar reinforced composites, PALF reinforced composites had a lesser density. Because of the decreased density of PALF, the composite thickness had to be increased in order to maintain the same weight fraction of reinforces in hybrid composites. Though [KPK]s and [KKP]s had the same weight, there was a slight difference in density between Kevlar fibres and PALF. This is owing to the fact that the amount of matrix immersed by samples varied depending on the stacking sequence and voids created by trapped air during the samples manufacturing. It could also be suggested that the void content and inconsistency mixing of the UP during the laminated process, affected the final substance and compactness of the products. Amirkhosravi et al. mentioned that the vacuum bagging technique that assisted with high magnetic compressive can reduce up to 70% reduction in void volume fraction [30]. This void percentage of the composite can be controlled by adjusting the moulding parameters and led to improve the laminates mechanical properties, and thermal insulation [31].

#### 3.1.2. Moisture Characteristics of PALF/Kevlar-Reinforced UP Hybrid Composites

The percentage of the moisture absorption of PALF/Kevlar reinforced UP hybrid composites with various laminated configurations ([PKP]s, [PPK]s, [KPP]s, [KKP]s, [PPP]s and [KKK]s) against square root of the moisture-exposure time is plotted in Figure 4. Based on Figure 4, all specimens absorbed moisture drastically at the beginning of the moisture absorption test. Then, it reached the saturation state at the highest peak of the curve before the rate of moisture absorption uptake reached to a constant rate, which meant there was no water absorption by the composite laminated. This behaviour has been observed in the previous studies [10]. Table 3 presents the moisture absorption characteristics of [PKP]s, [PPK]s, [KPP]s, [KKP]s, [PPP]s and [KKK]s.

From the results of the moisture absorption test shown in Table 3, the pure composite laminate [PPP]s showed the maximum moisture absorption of 17.76% and the pure [KKK]s showed the minimum moisture absorption of 6.49 %. Amongst all the hybrid composites, [PPK]s absorbed the highest percentage of moisture at 16.02%, whereas the [KKP]s and [KPK]s absorbed the minimum percentage of moisture at 14.18% and 14.17%, respectively as compared to the other stacking patterns. As can be seen from all the composite laminates, the percentage of moisture uptake increased with the volume fraction of PALF in the samples. It was clear that PALF reinforced UP composite [PPP]s showed the highest moisture absorption due to its hydrophilic in nature. The reinforced blends have a larger rise in water absorption, which can be attributed to the natural fibres’ water uptake capabilities. Natural fibres, in particular, have a high hydrophilic tendency due to the abundance of hydroxyl groups in their structure [32]. These hydroxyl groups help them connect with water molecules more easily. The hydrophilic character of the reinforcing fillers would invariably aid the reinforced blends’ affinity for moisture, which is thought to be the cause of the increased water absorption.

From Table 3, it was understood that the hybrid composite [KKP]s has less diffusion coefficient and permeability (2.36 × 10^−5^ mm^2^/s) than other hybrid composites [PKP]s, [PPK]s and [KPK]s which indicates that [KKP]s hybrid possessed good moisture resistance behaviour. However, with similar percentage of moisture absorption uptake, [KPK]s has better diffusion rate (3.09 × 10^−5^ mm^2^/s) which absorbed moisture faster than [KKP]s and other hybrid composite samples. By using PALF as second and outer layer as compared to other hybrid samples, had enabled the PALF composite to absorb a higher percentage of moisture than PU matrix and Kevlar fibre due to its hydrophilic characteristic. This observation was similar to the study conducted by Naveen et al. [33].

#### 3.1.3. Thickness Swelling Characteristics

The thickness swelling of PALF/Kevlar reinforced UP hybrid composites are presented in Figure 5 and the summary of the thickness swelling are tabulated in Table 4. Generally, the thickness swelling was affected by the PALF loading similar to the percentage of the water absorption. As the PALF content increased, the amount of water absorbed in the hybrid composite also increased as indicated by the increase of the hybrid composite thickness. From the result shown in Table 4, the highest TS_m_, was pure [PPP]s at 7.34%. As expected, PALF is a natural fiber with hydrophilic characteristic in nature. Amongst the hybrid composite laminates, [PPK]s exhibited the highest thickness swelling followed by [PKP]s with percentage of thickness swelling of 4.73% and 4.49%, respectively. However, the thickness swelling for [KPK]s and [KKP]s was 2.56% and 2.36%, respectively. The lowest thickness swelling yields by [KKK]s at 0.77 %. This can be explained by the hydrophilicity of the hybrid composite containing PALF which contain large amount of carbonyl and hydroxyl groups. The cellulose fibre possesses lumens at the central hollow and allow more water to be absorbed by capillary effect. The water have high tendency to penetrate through micro-gaps and partial debonding in the interface and cause the swelling of the hybrid composites [34]. Other than that, using natural fibre as the outer layer also contributed to high thickness swelling, as can be seen from the result, increasing the number PALF as the outer layer increased the thickness swelling.

### 3.2. Mechanical Properties of PALF/Kevlar-Reinforced Unsaturated Polyester Hybrid Composites

#### 3.2.1. Tensile Properties

The six samples of inter-ply PALF/Kevlar reinforced UP hybrid composites were loaded by using the universal testing machine in uniaxial tension till break up. In order to evaluate the Young’s modulus and tensile strength values, a typical stress-strain plot was obtained as shown in Figure 6. The [KKK]s composite curve which closely obeyed the Hooke’s law were initially a linear stress-strain plot that represented the deformation of each cell wall followed by sudden fracture of the samples [35]. However, [PPP]s and hybrid composites exhibited nonlinear stress-strain behaviour. Figure 11 shows the [KKK]s’ tensile properties outstanding performance than the other five composite configurations. However, the results also indicated that the tensile properties of hybrid composites were higher than the 6P. In general, [KKK]s composite elucidated a brittle failure, however, six plies of PALF laminates shown a pseudo-ductility behaviour. A similar finding was also reported by Wang et al. It indicated a combination between the high-ductility of synthetic fabric and low-ductility woven natural fibre, resulting a pseudo-ductility manner [36]. It could be suggested that by adding a woven Kevlar would reduce the nonlinearity of the remaining hybrid laminates. In addition, the fracture strain of hybrid composites showed a negative effect, whereas the [KPK]s sample was lowered by 4% as compared to the [KKK]s composites. In addition, all the hybrid composites mentioned above yielded almost an equal strain to failure values of about 2.7%.

Figure 7 illustrates the tensile strength and Young’s modulus of six configurations of PALF/Kevlar-reinforced UP hybrid composites. In this study, the maximum tensile strength and tensile modulus of the 6K composite plates were 224.6 MPa and 15.7 GPa, respectively which approximate 3.5 times to those of pure PALF reinforced UP composite. The tensile properties of the PALF/Kevlar reinforced UP hybrid composites were associated with the arrangement of the woven cloth. Hybridisation of the woven PALF fibre and Kevlar fabric resulted in an increase in the tensile strength and modulus as compared to those of the pure PALF composites. This showed that hybridisation of the laminate composite produced a novel material with intermediate tensile properties between the Kevlar and PALF woven fibres used alone [37]. Moreover, the mechanical behaviour of the inter-ply natural fibre-reinforced composites was lower due to the different nature of the hydrophobic matrix and the hydrophilic fibres, resulting in the reduction of adhesion and compatibility of the matrix-fibre interaction. To improve the tensile properties of natural fibre composites, the hybridisation with woven Kevlar technique should be used. The strength and modulus of [KKP]s and [KPK]s hybrid systems were greater than those of the [PPK]s and [PKP]s, proposing that the synthetic layers located at the outer surfaces of the laminate had improved the mechanical properties of the composite. A similar finding was also reported by Yahaya et al. [38]. The mechanical properties became higher when the high strength material was used in the outer plie’s specimen due to the better load transfer from the weak PALF fabric and consequently, the crack arrest was improved. As shown in Figure 6, PALF reinforced UP composites has a low failure strain, and thus this composite in a hybrid system could first reached failure. The catastrophic nature of the brittle failure of the PALF composite layer resulted in cracks at the PALF/Kevlar interface adjacent layer. As the cracks expanded, the cracks in the PALF fibres grew and formed dreadful failure clusters. When the failure clusters reached the critical point, the composite suddenly failed. Such improvement in the PALF-Kevlar-reinforced UP hybrid composites properties was mainly due to the high strength and modulus values of woven Kevlar than the PALF cloth inferior properties.

#### 3.2.2. Flexural Properties

The flexural test was conducted to determine the materials ability to withstand bending forces by measuring the flexural strength and flexural modulus. Figure 8 shows the flexural strength and flexural modulus of PALF/Kevlar reinforced UP hybrid composites. It showed that the [KPK]s highest flexural strength at 164.0 MPa followed by [KKP]s at 143.0 MPa. Also, the [PPP]s lowest flexural strength at 62.0 MPa. Furthermore, the [KKP]s highest flexural modulus at 12.6 GPa followed by [KPK]s at 8.1 GPa. The main results were similar to a previous study by Karimzadeh et al. on mechanical properties and moisture absorption characteristic of hybrid PALF/Glass fibre composites which had the highest flexural strength of 170.3 MPa for PGPG stacking sequence due to the increasing layer thickness made from similar materials had caused higher interlayer delamination [20]. By giving alternative stacking sequence, the thickness of layers made from similar materials reduce the crack propagation due to less interlayer delamination. Ismail et al. also mentioned that the outer layer of the hybrid composite laminates played an important role as the main load carrying member [39]. It controlled the bending strength and stiffness of the composites. Pure Kevlar fibre has the highest flexural strength than pure PALF fibre. By using the Kevlar fibre as the outer layer and arranged in an optimum stacking sequence, the bending strength of the hybrids would give the best performance.

#### 3.2.3. Impact Properties

The impact test was used to measure the impact resistance and impact strength in composite laminates before the sample failure occurred. Figure 9 shows the graph of the impact resistance and impact strength of PALF/Kevlar reinforced UP hybrid composites. From the results, [KKK]s had the highest impact resistance of 1.24 kJ/m. However, amongst the hybrid composite laminates, [KKP] showed the highest flexural resistance of 0.78 kJ/m followed by [KPK]s with a flexural strength of 0.68 kJ/m. The [PPK]s and [PKP]s exhibited flexural strength of 0.4 kJ/m and 0.33 kJ/m, respectively which were 78–83% times higher as compared to pure [PPP]s with flexural strength of 0.07 kJ/m. The performance of impact strength with different layering sequences of composite was similar to Kevlar highest impact resistance of 100 kJ/m^2^. For the hybrid composite, [KPK]s gave the highest impact strength of 65.5 kJ/m^2^ followed by [KPK]s of 55.3 kJ/m^2^. [PPK]s and [PKP]s exhibited impact strength of 36.6 kJ/m^2^ and 31.5 kJ/m^2^, respectively. Lastly, pure [PPP]s yielded the lowest impact strength of 6.4 kJ/m^2^. From the previous study conducted by Kartal and Demirer, the presence of only Kevlar fibre has resulted in the highest strength, as expected [40]. Whereas, for hybrid PALF/Kevlar, by increasing the fibre loading for Kevlar increased the impact strength of up to 91% than pure PALF fibre composite. Therefore, the impact strength also improved by using Kevlar as the outer layer as compared to PALF layer.

### 3.3. Scanning Electron Microscopy (SEM)

Figure 10 shows the scanning electron microscopy (SEM) images of PALF reinforced unsaturated polyester composites. The morphology of the surface was rough, with long fibre pull-out and fracture observable. It demonstrated PALF debonding from the matrix and fibre pull-out from the epoxy matrix. This indicated that the contact between the fibre and matrix was weak, resulting in poor adhesive bonding and a concentrated deformation zone [28]. The short and long fibre pull-out fractures in laminates indicated brittleness, whereas long fibre pull-out indicated pseudo-ductility [41].

Figure 11 illustrates the surface morphology for PALF/Kevlar hybrid composites. The presence of unsaturated polyester deposited along the PALF fibre demonstrated interfacial adhesion between the matrix and fibre, contributing to the composite sample’s good mechanical properties. After the tensile loading, the sample showed fibre fracture. This showed that fibre hybridisation had caused stress to be transmitted from the matrix to the PALF fibre prior to the sample failure. The hybrid laminate image revealed that the failure began on the external layer and spread to the inside layer, which was supported by the PALF layer’s extended fibre pull-out. Delamination was also detected between the hybrid interfaces, indicating that the Kevlar layers were more ductile. According to both failure modes, the laminates failed from the outside to the interior layer, and the stacking sequence has a favourable effect on the hybrid composites’ mechanical properties.

## 4. Conclusions

The effect of hybridisation on the mechanical properties of PALF/Kevlar UP hybrid composites was discussed. Recent research on PALF-reinforced polymers was highlighted, with a focus on moisture absorption, tensile, flexural, and impact properties. Several studies had looked into combining PALF with synthetic fibres other than Kevlar. From the findings, the hybridisation technique had improved the moisture absorption characteristics and mechanical qualities of the PALF composites. From this study, the stacking sequence of the PALF and Kevlar fibre laminates had a significant impact on these areas. The pure and hybrid composites’ moisture absorption behaviour were well expected. In comparison to the other stacking configurations, the hybrid composite with the alternating stacking pattern of [KKP] s and [KPK] s had the lowest percentage of moisture uptake at the saturated state (14.16% and 14.88%). Furthermore, the [KKP]s and [KPK]s stacking sequences increased the pure PALF composite’s flexural and tensile strengths more than the other stacking sequences. The hybridisation represented a pseudo-ductility created by combining high-ductility synthetic fabric with low-ductility woven natural fibre. In terms of impact strength and resistance, [KKP]s outperformed the composite laminates. The SEM scans revealed that the hybrid composite had greater interfacial adhesion between the fibres and matrix than pure PALF composite. The high strength and modulus values of woven Kevlar, rather than the inferior qualities of the PALF fibre, accounted for the improvement in PALF-Kevlar-reinforced UP hybrid composite properties. As a result, PALF/Kevlar hybrid composites have potentials in various applications.

## Figures and Tables

**Figure 1 polymers-14-02468-f001:**
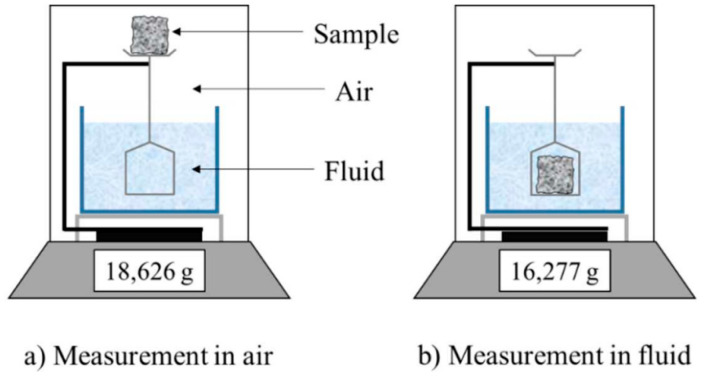
Scheme of Archimedes’ method for density measurement (adapted with permission, Elsevier, 2019).

**Figure 2 polymers-14-02468-f002:**
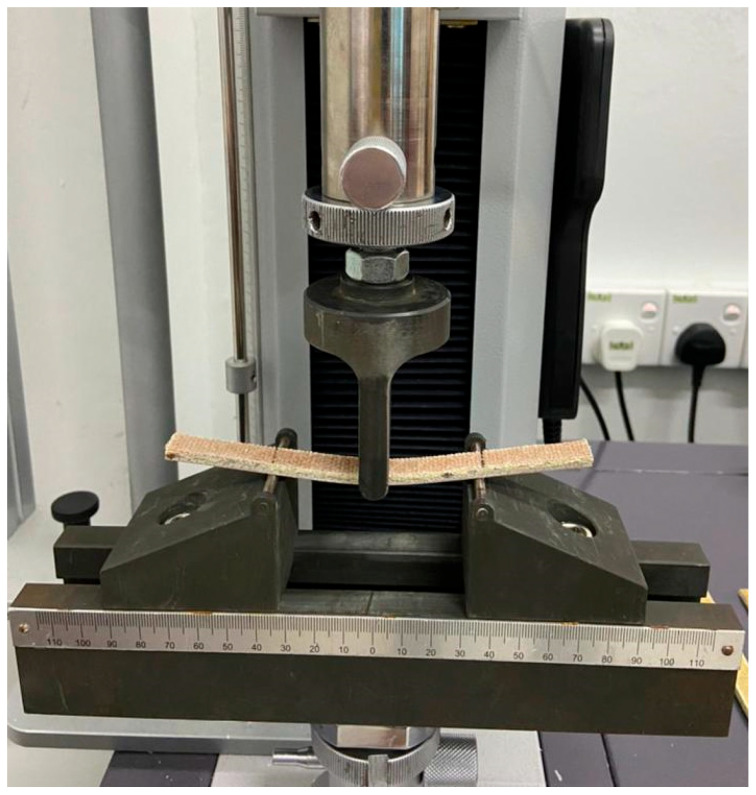
Photos of flexural loading and samples during flexural test.

**Figure 3 polymers-14-02468-f003:**
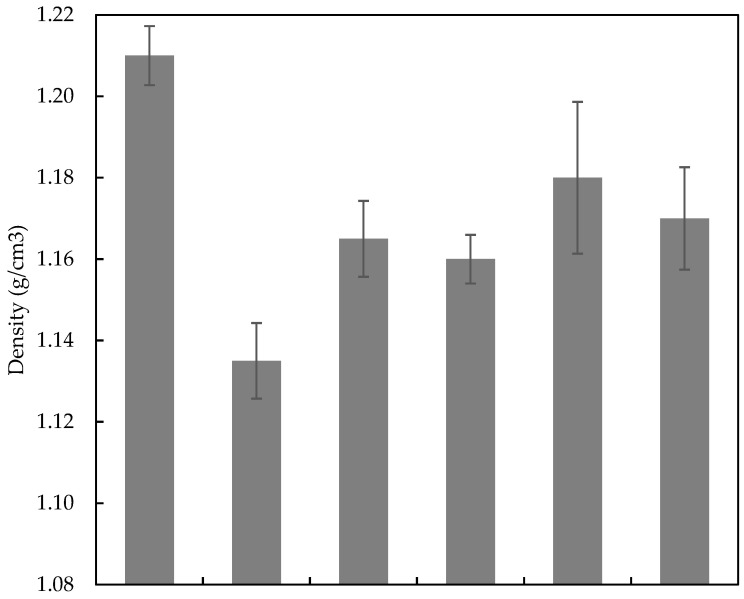
Density plots for PALF/Kevlar-reinforced UP hybrid composite laminates with different stacking sequences.

**Figure 4 polymers-14-02468-f004:**
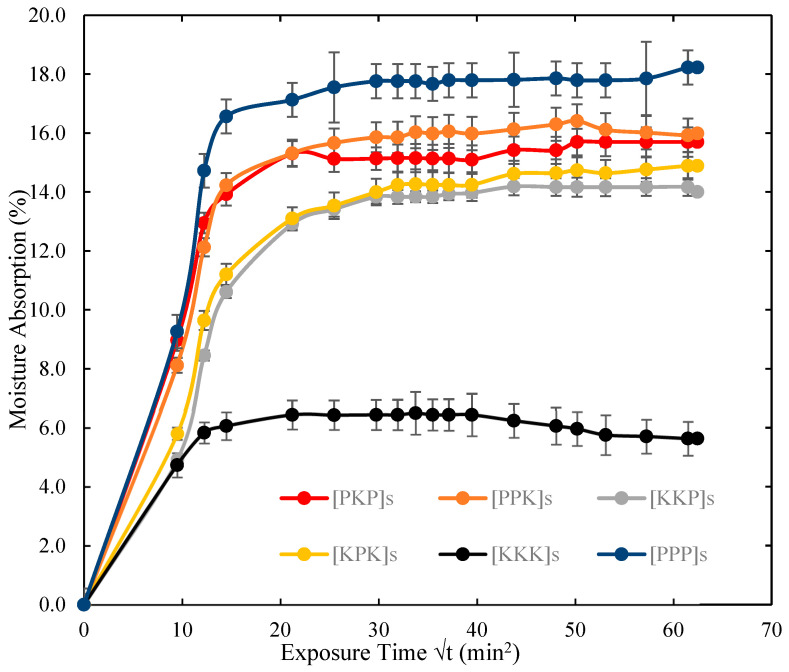
Moisture absorption characteristics of PALF/Kevlar-reinforced UP hybrid composite laminates with various stacking sequences.

**Figure 5 polymers-14-02468-f005:**
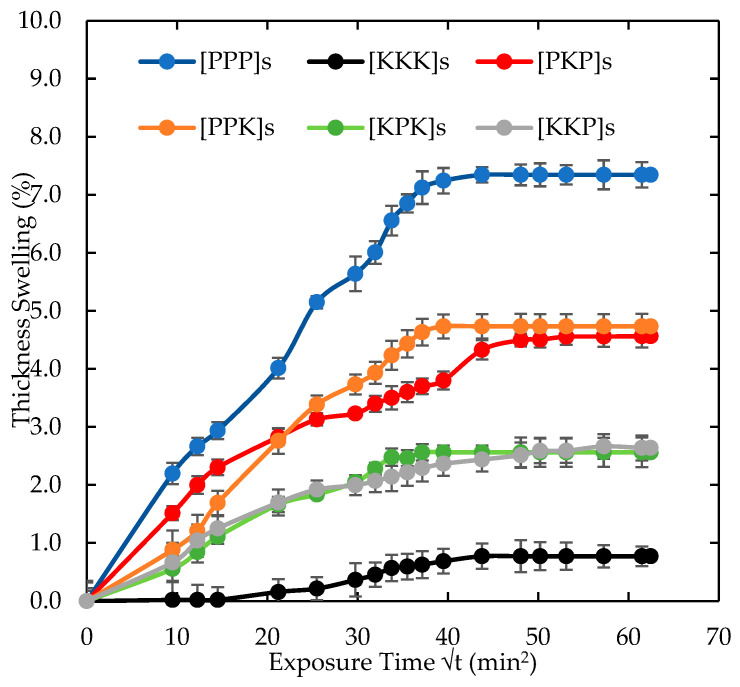
Thickness swelling of PALF/Kevlar-reinforced UP hybrid composite laminates with different stacking sequences.

**Figure 6 polymers-14-02468-f006:**
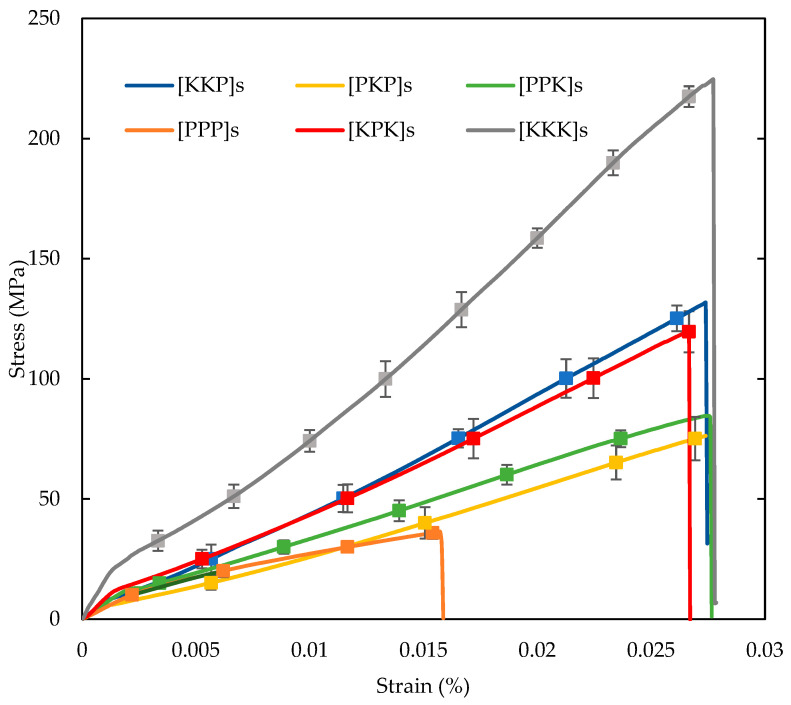
Tensile stress–strain curve of PALF/Kevlar-reinforced UP hybrid composite laminates with various stacking sequences.

**Figure 7 polymers-14-02468-f007:**
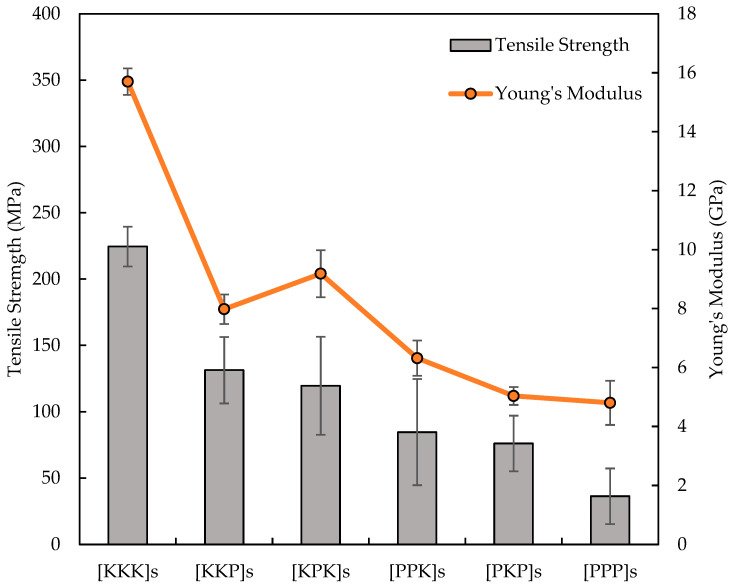
Tensile strength and Young’s modulus of PALF/Kevlar-reinforced UP hybrid composite laminates with different stacking sequences.

**Figure 8 polymers-14-02468-f008:**
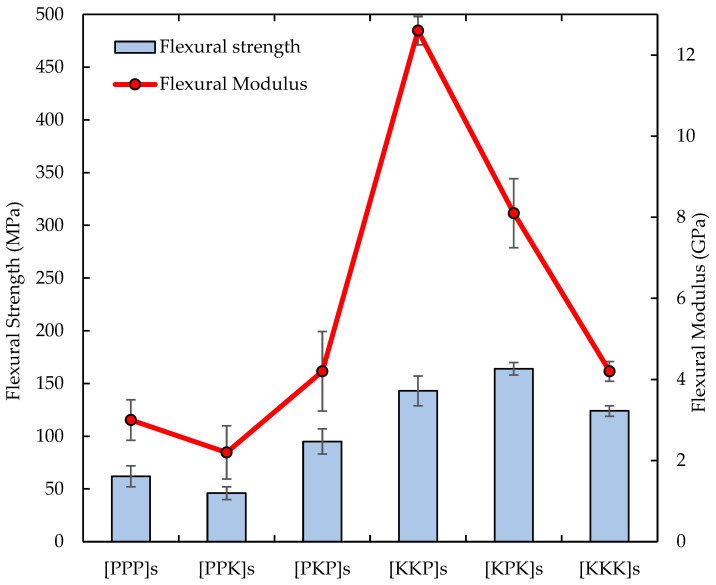
The flexural strength and flexural modulus of PALF/Kevlar-reinforced UP hybrid composite laminates with different stacking sequences.

**Figure 9 polymers-14-02468-f009:**
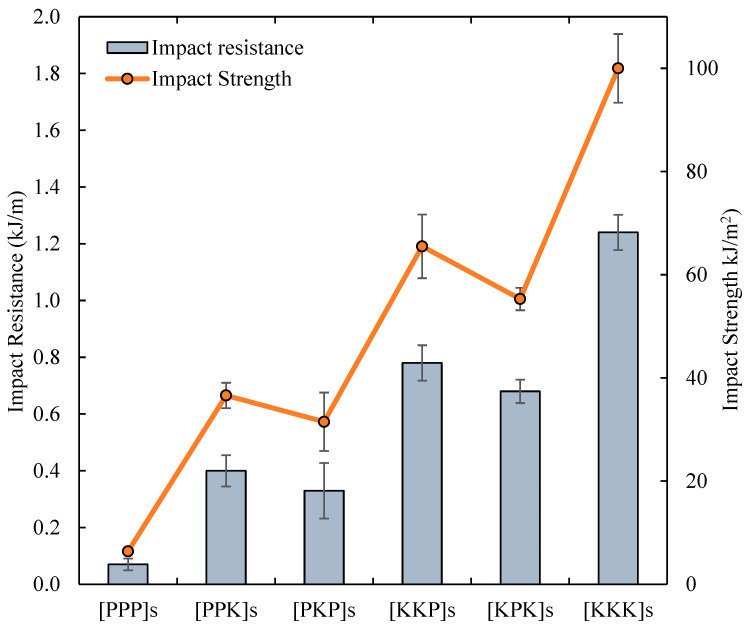
The impact resistance and impact strength of PALF/Kevlar-reinforced UP hybrid composite laminates with different stacking sequences.

**Figure 10 polymers-14-02468-f010:**
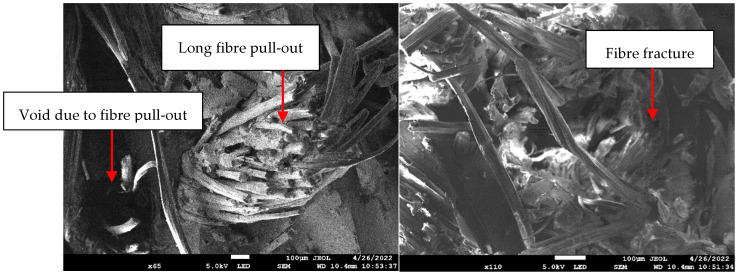
Scanning electron microscopy images of the PALF composites under tensile loading.

**Figure 11 polymers-14-02468-f011:**
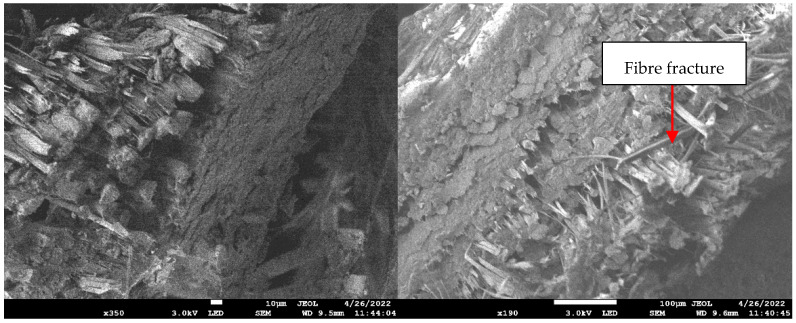
Scanning electron microscopy images of the PALF/Kevlar hybrid composite laminates under tensile loading.

**Table 1 polymers-14-02468-t001:** Research works reported on PALF hybrid composite laminates.

Synthetic	Matrix	Hybridization Technique	Configuration	Experimental Evaluation	Research Findings	References
Glass	Epoxy	Interply	BGB, BGP, PGP	Flexural strength = 105.87 MPa Flexural modulus = 7.613 GPa	For both banana and PALF hybrid composites, the optimal fiber volume composition is 40%.	[19]
Glass	Epoxy	Interply	4P, PGPG, PGGP, GPPG	Tensile strength = 119.21 MPa Tensile modulus = 5.07 GPa Flexural strength = 170.73 MPa Flexural modulus = 3.27 GPa	The PGPG stacking sequence increased the pure PALF composite’s flexural and tensile strengths by 119% and 153%, respectively, compared to the other stacking sequences.	[20]
Glass	Epoxy	Interply	PALF PALF+glass Coir Coir+glass	Tensile strength = 52 MPa Tensile modulus = 2.1 GPa Flexural strength = 145 MPa Flexural modulus = 7 GPa Impact strength = 7.5 kJ/m^2^	The qualities of a composite created by combining chemically treated cellulosic and glass fibers in an optimal volume ratio outperform those of a single glass fiber-reinforced material.	[21]
Basalt	Epoxy	Interply	2B_0_2P_0_2B_0_ 2B_0_2P_30_2B_0_ 2B_0_2P_45_2B_0_ 2B_0_2P_60_2B_0_ 2B_0_2P_90_2B_0_	Tensile strength = 262 MPa Tensile modulus = 10.8 GPa Flexural strength = 292 MPa Flexural modulus = 21.66 GPa	The storage modulus and loss tangent, as well as other mechanical parameters examined, will be greatly influenced by changes in fiber orientations.	[22]
Glass	Epoxy	Interply	SGBP, SGB, SGP	Flexural strength = 112.3 MPa Flexural modulus = 8.56 GPa	Increases in fiber length have always had a positive impact on mechanical and thermal qualities.	[23]
Carbon	Epoxy	Interply	PPPP-untreated PPPP-treated PCCP CPPC	Tensile strength = 187.67 MPa Tensile modulus = 7.87 GPa Flexural strength = 131.5 MPa Flexural modulus = 4.82 GPa	The laminate’s overall tensile characteristics were aided by the carbon ply inside (PCCP).	[23]
Glass	Polypropylene	Interply	GGG, GPG, PGP, PPP	Tensile strength = 58.26 MPa Tensile modulus = 6.13 GPa Flexural strength = 62.23 MPa Flexural modulus = 2.39 GPa Impact strength = 25 kJ/m^2^	The enhancement of the mechanical properties can be attained through hybridization.	[24]
Glass	Epoxy	Intraply	PGE	Flexural strength = 114.4 MPa	The composite has the best flexural strength when the PALF content volume is 20%, the glass fiber content volume is 20%, and the fiber length is 25 mm.	[25]
Glass	Epoxy	Interply	G, GP, P	-	The results suggest that PALF with 10% PALF and woven glass fiber with 20% PALF have the best tensile strength and stiffness.	[26]
Carbon	Epoxy	-	-	-	In the hybrid composite, increasing the CF content up to 30% increases the tensile and flexural strengths.	[27]
Carbon	Epoxy	Interply	PCP, CPC, PCPC, CPCP	Impact resistance = 1.64 kJ/m Impact strength = 100.29 kJ/m^2^	The addition of carbon fiber to the hybrid composite boosted the composite’s impact resistance and impact strength, making it stronger and more resistant to breaking.	[28]

**Table 2 polymers-14-02468-t002:** Composition of PALF and Kevlar laminate schemes in the hybrid composite.

Specimen	Laminate Scheme	Laminate Sequence	No. of PALF Woven Fabrics	No. of Kevlar Woven Fabrics	Thickness (mm)
[KKK]s		K + K + K + K + K + K	-	6	3.75
[KKP]s	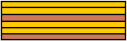	K + K + P + K + K + P	2	4	4.75
[KPK]s		K + P + K + K + P + K	2	4	6.35
[PPK]s	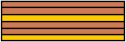	P + P + K + P + P + K	4	2	5.00
[PKP]s	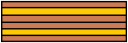	P + K + P + P + K + P	4	2	6.75
[PPP]s	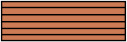	P + P + P + P + P + P	6	-	5.25

**Table 3 polymers-14-02468-t003:** Summary of moisture absorption characteristics of PALF/Kevlar-reinforced UP hybrid composites.

Hybrid Laminate	Maximum Moisture Uptake M_max_ (%)	Moisture Uptake at Infinity Time (%)	Diffusivity Coefficient (×10^−5^ mm^2^/s)	T M_max_ (min)
[KKK]s	6.49	5.63	2.53	1140
[KKP]s	14.18	14.16	2.36	1560
[KPK]s	14.17	14.88	3.09	2310
[PKP]s	15.30	15.69	5.71	450
[PPK]s	16.02	15.98	6.38	1140
[PPP]s	17.76	18.23	8.08	885

**Table 4 polymers-14-02468-t004:** Summary of thickness swelling of PALF/Kevlar-reinforced UP hybrid composites.

	Maximum Moisture Uptake, Mmax (%)	Moisture at Infinity Time, M∞ (%)	Diffusivity Coefficient, D (×10^−5^ mm^2^/s)	Maximum Thickness Swelling, TSm (%)
[PKP]s	15.30	15.69	5.71	4.49
[PPK]s	16.02	15.98	6.38	4.73
[KKP]s	14.18	14.16	2.36	2.36
[KPK]s	14.17	14.88	3.09	2.56
[KKK]s	6.49	5.63	2.53	0.77
[PPP]s	17.76	18.23	8.08	7.34

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
