# Peer review of "Mechanical Properties of PALF/Kevlar-Reinforced Unsaturated Polyester Hybrid Composite Laminates"

_polymers, 2022, doi:10.3390/polym14122468_

Round 1

Reviewer 1 Report

Title: “Mechanical Properties of Palf/Kevlar reinforced Unsaturated Polyester Hybrid Composites Laminates”

This study aims to determine the physical and mechanical properties of PALF/Kevlar reinforced Unsaturated Polyester hybrid composites. In particular, the PALF/Kevlar hybrid composites were fabricated by using hand-layup method using Unsaturated Polyester as matrix and were laid up to various laminated configuration such as [PKP]s, [PPK]s, [KPP]s, [KKP]s, [PPP]s and [KKK]s. First, the density of PALF/Kevlar reinforced Unsaturated Polyester were evaluated. The highest density result obtained from [KKK]s, however, the density of hybrid composites was closely indistinguishable. Moreover, moisture absorption behavior and its effects on the PALF/Kevlar reinforced Unsaturated Polyester were investigated. The water absorption studies shows that the hybridization between PALF and Kevlar all specimens absorb moisture drastically at the beginning of the moisture absorption test and the percentage of moisture uptake increased with the volume fraction of PALF in the samples. Tensile test indicated that all specimens exhibit nonlinear stress-strain behavior and shown a pseudo-ductility behavior and [KKP]s and [KPK]s are the hybrid composites that have highest tensile strength and modulus. Flexural test was showed that the highest flexural strength was [KPK]s with value 164.0 MPa and flexural modulus was [KKP]s with value of 12.6 GPa. In terms of impact strength and resistance, [KKP]s beat composite laminates. The hybrid composite demonstrated stronger interfacial adhesion between the fibres and the matrix than the pure PALF composite, according to SEM scans.

General comment: Although the topic of the work is interesting the work should be revised to enhance its quality and impact. In particular, some more figures related to the analysed structures should be added to the main text to allow the interested readers to better understand the work. A better explanation of models should be provided together with a more correct assessment of errors. In addition, also the quality of the language should be enhanced in some parts of the work.

Some detailed comment:

Table 1. Research works reported on PALF hybrid comosite laminates.

*) Although this table is interesting, it contains several information, so it should be better explained to the interested readers. In particular, a better caption should be inserted in the main text (please correct the typos). Furthermore, an image of the configuration should be inserted to allow also a visual understanding.

Lines: “2.2 Fabrication of Sample Hybrid Composite 109

The specimens fabricated are interply non-hybrid and hybrid PALF/Kevlar rein- 110

forced unsaturated polyester hybrid composites. Six different types of composites, includ- 111

ing two non-hybrid laminate and four hybrid laminates, were produced. After applying 112

mould release wax to the glass mould surface, peel-ply and other fibres were inserted one 113

by one in the desired stacking sequences ([PKP]s, [PPK]s, [KPP]s, [KKP]s, [PPP]s and 114

[KKK]s). Excess epoxy from roller action drips down the edges of the specimen and fre- 115

quently adheres to the glass mould surface. The purpose of peel ply is to make it easy to 116 etc,”

and

Table 2. Composition of PALF and Kevlar in the hybrid composite

*) Please add a least an image and a scheme for each composite. Indeed, it is not clear what are the main directions in all the composite materials.

Lines. “2.3.1. Density measurement 130

The Archimedean principle was used to calculate the density measurement. The 131

solid condition of the finally cured composite sample was weighed in air, then weighed 132

again in a known density liquid (alcohol). The density estimated from measured values 133

in g/cm3”

*) Although this measurement is based on an elementary physical principle a scheme of the used devices is needed to better understand the process. Please add a figure. Also the assessment of the errors should be added and discussed in the “Results” section.

Lines: “2.3.2. Water absorption characteristic 136

The ASTM D570 standard was used to measure water absorption characteristics. The 137

sample was dried for two hours at 80 °C in a circulation oven before getting the weight of 138

composites (CMH Ltd., Lancing, UK). In distilled water, five replicates of composite cou- 139

pons with dimensions of 20 x 20 x 3 mm were soaked. The samples were taken out of the 140

damp environment and dried with a microfibre cloth on all exposed surfaces. The mois- 141

ture uptake data were recorded using weight balance EMS 300-3 regularly at every 2 hours 142

of water immersion. The moisture absorption characteristic was evaluated using the fol- 143

lowing Equation (3.1) [29]: ,etc..

Wt is the sample’s weight at a recorded immersion time, and W0 is the weight of the 148

dried sample. The kinetic and diffusion mechanism was evaluated based on Fickian’s the- 149

ory. Its relationship is: 150

151

log [?∞?] = log⁡(? + ?og(? (3.2) 152

153

where Mt and M∞ are the water absorption at time, t and saturation point, respec- 154

tively. k and n are constants. 155

Diffusion coefficient, D or the ability of moisture to infiltrate into the molecule of a 156

composite, is a primary parameter in Fick's model. The D value is obtained from the initial

linear portion of the moisture absorption percentage versus the square root of the time 158

curve. The one-dimensional diffusion coefficient, D, can be determined from the following 159

equation: 160

161

?= ?[4?∞]2 [√?2 2− −?√?1]2 (3.3) 162

163

where h is plate thickness, M2, M1 are moisture content at time t1 and t2, respectively. 164

*) This paragraph should be improved by adding all the needed details about the used models. Could the authors better explain all the used Equations for the Fick model of diffusion. The composite is a 3D solid, why to use only the 1D model ? What are the main assumptions and simplifications of the model, what are the errors ? Please explain in a better way.

Figure 3. Moisture absorption behavior of PALF/Kevlar reinforced UP hybrid composites

Figure 4. Thickness swelling of PALF/Kevlar reinforced UP hybrid composites.

Figure 5. A typical tensile stress-strain curve of PALF/Kevlar reinforced UP hybrid composites.

*) The errors should be added for each measurement and comments in the main text.

Figure 6. A typical tensile stress-strain curve of PALF/Kevlar-reinforced UP hybrid composites.

*) The caption of this figures is the same of the figure5, please correct.

*) The order of the list of layers in composites “PPPPPP PKPPKP PPKKPP KPKKPK KKPPKK KKKKKK” is not totally clear. Could the authors provide a better motivation for that ? Is it intentional this order in the plot ? Please explain better.

Figure 7: [PPP]s [PPK]s [PKP]s [KKP]s [KPK]s [KKK]s.

*) See the previous comment. In addition, this list order seems to be different with respect to the previous one .. Could the authors better explain these points ? The caption of the figure should be improved.

Figure 8. The impact resistance and impact strength of PALF/Kevlar reinforced UP hybrid composites.

*) This caption should be improved.

Figure 9. Scanning electron microscopy images of the PALF composites

Figure 10. Scanning electron microscopy images of the PALF/Kevlar hybrid composites

*) These images are interesting, but they should be better commented and all the relevant features of each composite material should be presented in all details. Please add at least an image (well commented) for each investigated structure.

4. Conclusion 409

The effect of the hybridization of PALF/Kevlar unsaturated polyester hybrid compo- 410

sites various stacking sequences was studied. Laminates of [PKP]s, [PPK]s, [KPP]s, 411

[KKP]s, [PPP]s and [KKK]s was examined on the moisture absorption characteristics, ten- 412

sile, flexural and impact properties. The hybridization technique improved the moisture 413

absorption characteristics and mechanical qualities of the PALF composites, according to 414

the findings. The stacking sequence of the PALF and Kevlar fibre laminates had a signifi- 415

cant impact on these areas. The pure and hybrid composites' moisture absorption behav- 416

iour was well expected. When compared to the other stacking configurations, the hybrid 417

composite with the alternating stacking pattern of [KKP]s and [KPK]s had the lowest per- 418

centage of moisture uptake at the saturated state (14.16 and 14.88 percent). Furthermore, 419

the [KKP]s and [KPK]s stacking sequences increased the pure PALF composite's flexural 420

and tensile strengths more than the other stacking sequences. [KKP]s outperforms com- 421

posite laminates in terms of impact strength and resistance. SEM scans revealed that the 422

hybrid composite had greater interfacial adhesion between the fibres and the matrix than 423

the pure PALF composite. 424

*) This section should be improved: a “Discussion“ section should be added to discuss in details the value of the novel findings claimed in this work with respect to the current state of the art. Please rework.

Reviewer 2 Report

The strength properties of polymer composites are still topical and give a wide field for research. Optimizing the properties is important for both the matrix and the fibers. The selection of appropriate fibers, their length and quantity has a significant impact on all mechanical parameters. The work is interesting and constitutes a good background for further research on the optimization of the mechanical properties of composites. The authors demonstrated the good technique of the researcher. The experiment was planned correctly. The introduction and the literature review provide a good background to the problem under consideration. The literature is up to date. The markings in table 2 are a bit hard to understand. The figures are correctly described and will complement the information contained in the main text well. Final conclusions are spelled correctly.

Author Response

Thank you so much for your comments and suggestions. Table 2 has been improved.

Reviewer 3 Report

This work is devoted to the study of the mechanical properties of laminates of unsaturated polyester hybrid composites reinforced with Palf/Kevlar. This work is written in an understandable language, the results are presented clearly and clearly, and the relevance is not in doubt. Table 1 is a plus, thanks to which it is easier to understand what has already been done in this direction and to outline the further path of research. Despite the many advantages of this work, there are some points that it is desirable to improve:
1. All abbreviations should preferably be given before the "introduction" paragraph.
2. In many points of the study, there is a lack of comparison of the obtained characteristics with those known from the literature.
3. It would be a good addition to this work to study the thermal stability of the resulting composites.
4. In addition, some other physicochemical analyzes could improve this work, and their results would help to identify the relationship with mechanical and physical characteristics.
5. Scanning electron microscopy data should preferably be given for all samples and compared both with each other and with literature data.
6. In many points of the study, comparison with literature data will help to draw some more voluminous conclusion and understand trends for this process.

Round 2

Reviewer 1 Report

Title: “Mechanical Properties of Palf/Kevlar reinforced Unsaturated Polyester Hybrid Composites Laminates”

This study aims to determine the physical and mechanical properties of PALF/Kevlar reinforced Unsaturated Polyester hybrid composites. In particular, the PALF/Kevlar hybrid composites were fabricated by using hand-layup method using Unsaturated Polyester as matrix and were laid up to various laminated configuration such as [PKP]s, [PPK]s, [KPP]s, [KKP]s, [PPP]s and [KKK]s. First, the density of PALF/Kevlar reinforced Unsaturated Polyester were evaluated. The highest density result obtained from [KKK]s, however, the density of hybrid composites was closely indistinguishable. Moreover, moisture absorption behavior and its effects on the PALF/Kevlar reinforced Unsaturated Polyester were investigated. The water absorption studies shows that the hybridization between PALF and Kevlar all specimens absorb moisture drastically at the beginning of the moisture absorption test and the percentage of moisture uptake increased with the volume fraction of PALF in the samples. Tensile test indicated that all specimens exhibit nonlinear stress-strain behavior and shown a pseudo-ductility behavior and [KKP]s and [KPK]s are the hybrid composites that have highest tensile strength and modulus. Flexural test was showed that the highest flexural strength was [KPK]s with value 164.0 MPa and flexural modulus was [KKP]s with value of 12.6 GPa. In terms of impact strength and resistance, [KKP]s beat composite laminates. The hybrid composite demonstrated stronger interfacial adhesion between the fibres and the matrix than the pure PALF composite, according to SEM scans.

General comment: Although the authors partially revised their work, some important major issues should be reworked to enhance its quality. In particular, experimental plots should be provided with errors, as standard in all scientific fields, together with a detailed description of the number of specimens and of the experimental procedure. A statistical treatment of the errors should be also provided. In addition, a detailed description of the used model (1D Fick’s law) should also provided within the main text of the manuscript. Unfortunately, without these changes the work is not suitable to be published in a high quality journal.

Some detailed missed comment:

Point 4: Lines: “2.3.2. Water absorption characteristic 136

The ASTM D570 standard was used to measure water absorption characteristics. The 137

sample was dried for two hours at 80 °C in a circulation oven before getting the weight of 138

composites (CMH Ltd., Lancing, UK). In distilled water, five replicates of composite cou- 139

pons with dimensions of 20 x 20 x 3 mm were soaked. The samples were taken out of the 140

damp environment and dried with a microfibre cloth on all exposed surfaces. The mois- 141

ture uptake data were recorded using weight balance EMS 300-3 regularly at every 2 hours 142

of water immersion. The moisture absorption characteristic was evaluated using the fol- 143

lowing Equation (3.1) [29]: ,etc..

Wt is the sample’s weight at a recorded immersion time, and W0 is the weight of the 148

dried sample. The kinetic and diffusion mechanism was evaluated based on Fickian’s the- 149

ory. Its relationship is: 150

151

log [??] = log(? + ?og(? (3.2) 152

153

where Mt and M∞ are the water absorption at time, t and saturation point, respec- 154

tively. k and n are constants. 155

Diffusion coefficient, D or the ability of moisture to infiltrate into the molecule of a 156

composite, is a primary parameter in Fick's model. The D value is obtained from the initial

linear portion of the moisture absorption percentage versus the square root of the time 158

curve. The one-dimensional diffusion coefficient, D, can be determined from the following 159

equation: 160

161

?= ?[4?∞]2 [√?2 2− −??1]2 (3.3) 162

163

where h is plate thickness, M2, M1 are moisture content at time t1 and t2, respectively. 164

*) This paragraph should be improved by adding all the needed details about the used models. Could the authors better explain all the used Equations for the Fick model of diffusion. The composite is a 3D solid, why to use only the 1D model ? What are the main assumptions and simplifications of the model, what are the errors ? Please explain in a better way. The following author’s response “Response 4: Thanks for the comments, this section need to be reworked. However, we need extra time to reworked this section and will improved in the manuscript.” is very unsatisfactory. Please rework.

Point 5:

Figure 3. Moisture absorption behavior of PALF/Kevlar reinforced UP hybrid composites

Figure 4. Thickness swelling of PALF/Kevlar reinforced UP hybrid composites.

Figure 5. A typical tensile stress-strain curve of PALF/Kevlar reinforced UP hybrid composites.

*) The errors should be added for each measurement and comments in the main text. The following response of the authors “Response 5: In these figures, the errors are not necessary.” is very unsatisfactory. I addition, the errors are standard for every measure in all the scientific fields, so it is definitely not clear why “ the errors are not necessary”. The authors should, for each plot, explicitly provide the number of specimens used and a statistical treatment of the error. This is important to show to the interested readers that all their claims are supported by experimental data. Please rework and improve.

Round 3

Reviewer 1 Report

It seems that the authors revised their main text according to the comments of this reviewer. Nevertheless, the style of the manuscript and the quality of the images and captions should be furhter improved. In particular, all plots should show also the errors and the captions should describe in details the number of specimnes used for each specific test.

Author Response

Comments:

It seems that the authors revised their main text according to the comments of this reviewer. Nevertheless, the style of the manuscript and the quality of the images and captions should be furhter improved. In particular, all plots should show also the errors and the captions should describe in details the number of specimnes used for each specific test.

Response:

Thank you so much for your comments and suggestions. The captions have been improved, all errors have been added to all plots and the number of specimens used in every experiment was included in subsection 2.3. For your information, we are currently proofreading this manuscript and it takes several days. We will submit it again once the proofreading is finished. Thank you.